# Advancements in Allergen Immunotherapy for the Treatment of Atopic Dermatitis

**DOI:** 10.3390/ijms25021316

**Published:** 2024-01-21

**Authors:** Bei-Cyuan Guo, Kang-Hsi Wu, Chun-Yu Chen, Wen-Ya Lin, Yu-Jun Chang, Mao-Jen Lin, Han-Ping Wu

**Affiliations:** 1Department of Pediatrics, National Cheng Kung University Hospital, College of Medicine, National Cheng Kung University, Tainan 70403, Taiwan; gbc628@gmail.com; 2Department of Pediatrics, Chung Shan Medical University Hospital, Taichung 40201, Taiwan; cshy1903@gmail.com; 3School of Medicine, Chung Shan Medical University, Taichung 40201, Taiwan; 4Department of Emergency Medicine, Tungs’ Taichung Metro Harbor Hospital, Taichung 43503, Taiwan; yoyo116984@gmail.com; 5Department of Nursing, Jen-Teh Junior College of Medicine, Nursing and Management, Miaoli 35664, Taiwan; 6Department of Pediatrics, Taichung Veteran General Hospital, Taichung 43503, Taiwan; wylin002@gmail.com; 7Laboratory of Epidemiology and Biostastics, Changhua Christian Hospital, Changhua 50006, Taiwan; 83686@cch.org.tw; 8Division of Cardiology, Department of Medicine, Taichung Tzu Chi Hospital, The Buddhist Tzu Chi Medical Foundation, Taichung 42743, Taiwan; 9Department of Medicine, College of Medicine, Tzu Chi University, Hualien 97002, Taiwan; 10College of Medicine, Chang Gung University, Taoyuan 33302, Taiwan; 11Department of Pediatrics, Chiayi Chang Gung Memorial Hospital, Chiayi 61363, Taiwan

**Keywords:** atopic dermatitis, eczema, allergen immunotherapy, treatment

## Abstract

Atopic dermatitis (AD) is a chronic inflammatory skin condition that affects individuals of all age groups, manifesting as a spectrum of symptoms varying from mild to severe. Allergen immunotherapy (AIT) involves the administration of allergen extracts and has emerged as a potential treatment strategy for modifying immune responses. Its pathogenesis involves epidermal barrier dysfunction, microbiome imbalance, immune dysregulation, and environmental factors. Existing treatment strategies encompass topical steroids to systemic agents, while AIT is under investigation as a potential immune-modifying alternative. Several studies have shown reductions in the severity scoring of atopic dermatitis (SCORAD) scores, daily rescue medication use, and visual analog scale (VAS) scores following AIT. Biomarker changes include increased IgG4 levels and decreased eosinophil counts. This review provides valuable insights for future research and clinical practice, exploring AIT as a viable option for the management of AD.

## 1. Introduction

Atopic dermatitis (AD), often referred to as atopic eczema, is a chronic inflammatory skin condition that primarily affects children but can also manifest in adults [1]. Many individuals experience severe suffering due to this disease. AD is frequently associated with elevated serum immunoglobulin E (IgE) levels and a personal or family history of type I allergies, asthma, and allergic rhinitis [2]. Several factors, such as skin barrier impairment, abnormal immune responses, genetic factors, and environmental agents, play critical roles in the pathogenesis of the disease [3]. In recent years, various treatment modalities have been investigated, including topical steroids, systemic immunomodulatory agents such as biologics and small molecules, and allergen immunotherapy (AIT). AIT, also known as allergen desensitization or hyposensitization, involves the administration of allergen extracts [4]. AIT can also modify the immune regulation of allergic responses [5]. Its goal is to help patients with allergic conditions develop clinical tolerance to allergens that typically trigger symptoms [6].

In this review, we aim to investigate clinical trials related to AIT in the treatment of AD. Herein, we aim to provide a comprehensive summary of the effectiveness and side effects within the context of AIT for AD. Furthermore, we discuss the potential future directions in this field.

## 2. Overview of Atopic Dermatitis

### 2.1. Epidemiology of AD

#### 2.1.1. Prevalence and Incidence of AD

A notable aspect of AD is its higher prevalence in young children in comparison to adults. The disease typically begins at the age of 5 years; however, approximately 26% of adults with AD report the onset of the disease in adulthood [1,7]. The prevalence of AD is estimated to be up to 20% among children and approximately 2–5% among adults [8,9,10]. In the Global Report on Atopic Dermatitis 2022, AD was reported to affect up to 20% of children and 10% of adults. The age-standardized prevalence per age group showed a bimodal curve, indicating a high prevalence of AD in young children that decreased as individuals reached adulthood; however, there was an upward trend in middle-aged and older populations [11]. AD has a slight female preponderance [12,13], but its onset occurs earlier in boys than in girls [14]. Moreover, the incidence of AD has increased by 2- to 3-fold over the past few decades in industrialized countries [15].

#### 2.1.2. Risk Factors Associated with AD

Risk factors for AD include filaggrin (FLG) gene mutations and a family history of atopic or allergic disease. Other factors that can contribute to AD include exposure to daycare, the level of parental education, socioeconomic status, the place of residence (rural vs. urban setting), smoking, the type of delivery during childbirth (vaginal vs. cesarean section), birth weight, breastfeeding, being overweight, exposure to hard water, contact with pets, and/or dust mites [16]. If either one of the parents has AD, the risk of AD development in their children increases threefold, and if both parents have AD, the risk increases fivefold [17]. Food allergies can also trigger episodes of AD [18]. AD has also been linked to a significant patient burden and a range of atopic comorbidities, such as asthma, hay fever, food allergies, and eosinophilic esophagitis. Additionally, non-atopic comorbidities, including infections, allergic contact dermatitis, anxiety, depression, suicidal thoughts, and cardiovascular diseases, have been associated with AD [19].

### 2.2. Pathogenesis of AD

The pathophysiology of AD is multifactorial and involves impaired epidermal barrier function, skin microbiome abnormalities, immune dysregulation, genetic factors, and environmental triggers of inflammation [20,21,22,23,24].

#### 2.2.1. Epidermal Barrier Dysfunction and Genetic Risk Factors

Research has estimated the heritability of AD to be approximately 75%, with a notable genetic risk factor being loss-of-function (LoF) mutations in the FLG gene located on chromosome 1q21.3 [25,26]. The occurrence of FLG LoF variants in children of African ancestry is lower than in children of European or Asian descent [27,28]. FLG deficiency results in a compromised stratum corneum lipid composition and organization, decreased levels of natural moisturizing factors, reduced skin hydration, and increased skin surface pH [29]. However, only a minority of patients with AD (approximately 10–40%) have loss-of-function mutations in the FLG gene [30]. These mutations are exclusively associated with early-onset AD and do not correlate with late childhood or adulthood onset of the condition [31]. In addition to the FLG gene, more than sixty other genes are implicated in AD [32]. The inflamed skin of patients with AD is characterized by lower quantities of certain antimicrobial peptides, including cathelicidins and defensins, which are essential components of the skin’s innate immune system [33]. The reduced presence of antimicrobial peptides, along with other abnormalities in the skin barrier, such as altered filaggrin levels, can render the inflamed skin more susceptible to penetration and infection by *Staphylococcus aureus* [33].

#### 2.2.2. Skin Microbiome

AD is associated with a disrupted microbiome. Notably, the interaction between microbes and immune cells residing in the skin appears to play a significant role in influencing the course of the disease [34]. A meta-analysis revealed that the prevalence of *S. aureus* among patients with AD is approximately 70% in lesioned skin and approximately 39% in non-lesioned or healthy skin. Furthermore, the prevalence increased with the severity of the disease [35]. A comprehensive analysis of AD flares revealed that patients with more severe disease symptoms often exhibited a higher predominance of *S. aureus*, while those with milder disease symptoms demonstrated a higher predominance of *Staphylococcus epidermidis* [36]. The reduced microbial diversity observed in patients with *S. aureus* skin colonization may contribute to the pathogenesis of AD in several ways, including barrier disruption and direct proinflammatory effects such as type 2 immune activation [37,38]. Furthermore, *S. aureus* generates enterotoxins (superantigens), which are recognized for their capacity to undermine the integrity of the skin barrier, leading to the amplification of type 2 inflammation [39]. Superantigens also downregulate the dermal synthesis of interferon-gamma and tumor necrosis factor-alpha, both of which play pivotal roles in cellular immunity against bacterial and viral infections [40]. Various substances, including alpha toxins, superantigens, toxic shock syndrome toxin 1, and enterotoxins produced by *S. aureus,* have been demonstrated to harm the skin barrier and/or provoke inflammation, ultimately contributing to the development of AD [41].

#### 2.2.3. Immunological Dysregulation and Inflammation

AD is characterized by defects in both innate and acquired immune responses, and a close relationship exists between immune dysregulation, epidermal barrier function, and microbiome dysbiosis [42]. In the context of the innate immune response, AD is linked to abnormalities in the signaling of certain pattern recognition receptors (PRRs), including the nucleotide-binding oligomerization domain (NOD), Toll-like receptors (TLR), and some soluble PRRs [43,44,45]. These abnormalities result in the reduced reactivity of NK lymphocytes, polymorphonuclears, and dendritic cells, as well as the decreased synthesis of antimicrobial peptides (AMPs) in the skin [45]. Furthermore, there is an overexpression of Th2 cytokines (interleukin (IL)-4, IL-5, thymic stromal lymphopoietin (TSLP), IL-13, and IL-31) during the acute phase [21,46], along with the upregulation of T-helper cell type 22 (Th22) cytokines, leading to the overexpression of IL-22 [47,48]. In chronic AD lesions, there is an intensification of T-helper cell type 2 (Th2) and Th22 responses along with the concurrent activation of the T-helper cell type 1 (Th1) axis that results in increased levels of interferon-gamma (IFN-γ), chemokine (C-X-C motif) ligand 9 (CXCL9), and chemokine (C-X-C motif) ligand 10 (CXCL10), rather than a complete shift to a Th1-only signature [48,49,50].

#### 2.2.4. Environmental Factors

Several environmental risk factors influence AD, some of which have a preventive effect, whereas others aggravate it. Protective factors include UV light exposure and the consumption of fresh fruits and fish in one’s diet during pregnancy and breastfeeding. Aggravating factors include lower temperature climates, urbanization, fast food consumption, delayed weaning, obesity, and pollution/tobacco smoke exposure [15,51,52]. Humidity can have various effects on atopic skin depending on the location and specific humidity levels [52]. Microbial exposure also influences the development of AD [53]. Daycare attendance in the first two years of life, the consumption of unpasteurized farm milk, pre- and postnatal exposure to farm animals, and dog exposure in early life are considered protective factors. On the other hand, high endotoxin exposure levels and/or exposure during the first year of life, as well as postnatal antibiotic exposure, have been identified as risk factors [15].

### 2.3. Clinical Manifestations of AD

#### 2.3.1. Common Features and Diagnosis of AD

The most common features of AD include pruritus, dry skin (xerosis), lichenification, a course influenced by emotional and/or environmental factors, flexural involvement, and early disease [54,55,56]. Adults have demonstrated a higher pooled prevalence of lichenification, erythroderma, disease course influenced by emotions and/or environmental factors, ichthyosis, palmar hyperlinearity, keratosis pilaris, nonspecific hand and foot dermatitis, dyshidrosis, prurigo nodularis, and papular lichenoid lesions [56,57]. In contrast, pediatric studies have reported a higher prevalence of dermatitis of the eyelid, auricular area, and ventral aspect of the wrist; exudative eczema; features resembling seborrheic dermatitis; and early disease onset (defined as onset before the age of 2 years) [56,57].

The Hanifin–Rajka criteria (HRC) represent the initial and extensively referenced diagnostic criteria for AD (Table 1) [58]. The criteria require the fulfillment of three of the four major criteria and three of twenty-three minor criteria. However, these criteria have limitations, which have led to the development of alternative sets. The United Kingdom Working Party and the American Academy of Dermatology proposed more streamlined criteria that are additionally applicable to the diagnosis of AD [2,59]. AD can be classified as acute, sub-acute, or chronic. The acute and subacute forms of AD typically manifest as intensely itchy red papules and vesicles with excoriations, accompanied by a serous exudate, whereas chronic AD is characterized by the presence of lichenified plaques and papules with excoriation [60]. According to the Japanese guidelines for AD, AD can be further categorized into mild, moderate, severe, and most severe based on the relative extent of the lesions in relation to the body surface area (Table 2) [61]. In laboratory findings, up to 80% of patients with AD exhibit elevated serum IgE levels, with the prevalence of eosinophilia being 25%, irrespective of concomitant food sensitization and disease severity [62].

#### 2.3.2. Severity of AD

The severity of AD is variable and can be recalcitrant to treatment [63]. Measuring disease activity is critical for the clinical management and monitoring of individual patients. Standardizing the measurement tools and harmonizing the outcome measures are vital for evidence-based practice. Considering the substantial burden of AD, various scoring systems have been established to measure disease severity [64]. The SCORAD (Severity Scoring Index of Atopic Dermatitis) assesses the extent (percentage of the area involved), intensity, and subjective symptoms (pruritus and sleep loss) on a scale from 0 to 103, defining three severity classes: mild (SCORAD < 25), moderate (25 ≤ SCORAD ≤ 50), and severe (SCORAD > 50) [65,66]. The EASI (Eczema Area and Severity Index) assesses the intensity of the lesions of AD in four different parts of the body (head and neck, upper limbs, trunk, and lower limbs) on a scale from 0 to 72 [67]. The SASSAD (Six Area Six Sign Atopic Dermatitis Atopic Score) involves evaluating six clinical features of disease intensity (erythema, exudation, excoriation, dryness, cracking, and lichenification) at six defined body sites (head and neck, arms, hands, trunk, legs, and feet) on a scale of 0 to 3, resulting in a maximum score of 108 [68].

## 3. Overview of AIT for the Treatment of Allergic Diseases

### 3.1. Mechanism of AIT

Allergen immunotherapy (AIT) is the only disease-modifying treatment available for allergic disorders [69]. Although AIT has been used to treat many atopic diseases, its precise mechanism of action remains unclear [70]. However, immunotherapy has been linked to a shift from T helper cell type-2 (Th2) immune responses, typically associated with the development of atopic conditions, to a more balanced state involving increased Th1 immune responses [71]. It is also linked to the generation of T regulatory cells (Treg), which produce anti-inflammatory cytokines, such as IL-10 and transforming growth factor β (TGF-β), leading to the early phase desensitization of effector cells (eosinophils, mast cells, and basophils) [72,73,74]. These allergen-specific Tregs also suppress Th2 cells by reducing the levels of allergen-specific IgE antibodies and increasing the levels of immunoglobulin G4 (IgG4), a non-inflammatory immunoglobulin isotype [75]. Cytokines such as IL-3, IL-4, IL-5, IL-9, and IL-13, which originate from Th2 cells, are essential for the survival, activation, and differentiation of mast cells, basophils, and eosinophils. Nevertheless, AIT is effective in the suppression of these cytokine axes [73,74].

### 3.2. Methods Employed in the Administration of AIT

AIT involves the gradual exposure of a patient to increasing doses of a specific allergen to reduce allergic and inflammatory responses, ultimately leading to a sustained decrease in allergic symptoms. AIT is a therapeutic vaccination used to treat IgE-mediated hypersensitivity to allergens [76]. Allergen extracts primarily consist of allergenic proteins derived from grass and tree pollen, dander, dust mites, insect venom, and mold [4,77,78]. *Dermatophagoides pteronyssinus* (Der p) and *Dermatophagoides farinae* (Der f) are recognized as the primary triggers of house dust mite (HDM) allergies globally. These two species share high homology and exhibit cross-reactivity [79]. HDMs are the most utilized allergens in AIT for AD [80,81].

The routes of AIT for food allergies are divided into several groups, including subcutaneous immunotherapy (SCIT), sublingual immunotherapy (SLIT), intradermal immunotherapy, and oral immunotherapy (OIT) [82,83,84,85]. The two primary forms of AIT in AD are administered either subcutaneously (SCIT) or sublingually (SLIT) [80,86]. In SCIT, the allergen extract is injected subcutaneously into the lateral or posterior middle portion of the arm [87], whereas SLIT involves the daily administration of antigen drops or tablets under the tongue [88].

AIT involves two main phases, including build-up and maintenance. In SCIT, the build-up phase consists of weekly injections, typically one to three injections per week. This phase starts with a very low allergen dose and gradually increases over a period of 3 to 6 months [77,89]. In contrast, in SCIT, patients receive a fixed allergen dose once daily during the maintenance phase. This administration can occur continuously throughout the year or pre-/co-seasonally, depending on the specific allergen triggering the symptoms and the type of allergen extract used [90]. Following this period, patients typically develop adequate tolerance to the allergen. This is associated with the induction of regulatory subsets of T and B cells, the generation of the IgG4 isotype, reduced inflammatory responses to allergens by effector cells in inflamed tissues, and the achievement of a maintenance dose [77,91,92,93]. During the maintenance phase, typically characterized by monthly injections through SCIT and three weekly administrations through SLIT, the treatment regimen is generally continued for a period ranging from 3 to 5 years [89,94,95].

### 3.3. Attributes of AIT

When comparing SLIT and SCIT, SLIT offers more convenience because it can be administered at home, making it suitable even for young children. Additionally, SLIT minimizes the use of healthcare resources and staff time and does not require specific expertise or facilities. The cost of the treatment extract is lower for SLIT; however, the overall cost, considering the increased use of healthcare resources, may be higher in SLIT [96]. Both SCIT and SLIT induce local and systemic reactions. The common local adverse effects of SCIT include swelling and redness at the injection site, whereas SLIT may cause oral itching and tingling. Systemic reactions affecting the entire body can range from anaphylaxis to asthma or urticaria and are generally uncommon in both methods, regardless of SCIT or SLIT [97,98,99]. Similar results regarding the adverse effects of AIT on AD have been noted [100].

### 3.4. Efficacy of AIT Treatment on Atopic Diseases

AIT has demonstrated significant clinical efficacy in the treatment of several allergic diseases, including allergic rhinitis, asthma, and allergic conjunctivitis [101,102,103,104]. According to the European Academy of Allergy and Clinical Immunology (EAACI) guideline, AIT can also be used for the treatment of hymenoptera venom allergy in both children and adults [105]. In a systematic review of animals, AIT was found to be a causative treatment for canine AD and demonstrated a satisfactory success rate with low adverse effects in both the short and long term [106]. AIT has been proven to be effective in several allergic diseases and can be regarded as a viable treatment option for AD [107].

## 4. AIT for the Treatment of AD

Many nonmedicinal and immunomodulatory agents are effective and safe for the treatment of AD. In national treatment guidelines, AD can be treated with nonpharmacologic interventions including bathing, showering and washing, emollients and moisturizers, and wet wrap therapy [108,109,110]; non-systemic topical treatments, including topical antihistamines, topical corticosteroids, topical calcineurin inhibitors (tacrolimus ointments), and topical Janus kinase inhibitors (Delgocitinib ointment) [108,109,110,111]; and systemic agents, including oral antihistamine, corticosteroid, cyclosporine, Azathioprine, Methotrexate, Mycophenolate mofetil, interferon-r, JAK1/JAK2 inhibitor, or biologics (Dupilumab or Tralokinumab), and phototherapy [108,109,111,112]. The use of antioxidants emerges as a potential therapeutic strategy in the treatment of AD, given the noted higher activity of catalase (CAT) as an enzymatic antioxidant in this condition [113]. The potential role of AIT in managing AD has been suggested because of its current status as an immune-modifying treatment for allergic diseases, making it a potentially effective option for AD treatment [107].

### 4.1. Outcomes of Recent Clinical Studies on AIT Treatment for AD

AIT was first introduced by Noon and Freeman in 1911 and has been successfully used for the treatment of several atopic diseases [4]. Numerous clinical studies have demonstrated the effectiveness of AIT in the treatment of AD. We gathered data from 10 studies spanning a decade to assess the effectiveness of AIT in AD treatment. This compilation included seven studies on SLIT and three on SCIT. The clinical outcomes of AIT in AD is discussed below, and a summary is provided in Table 3. Notably, all the aforementioned clinical studies demonstrated the safety of AIT, as only a few individuals experienced severe systemic reactions.

#### 4.1.1. The Efficacy of AIT in the Treatment of AD

AIT has shown promising outcomes in reducing the severity of AD. According to Yu et al., the mean SCORAD score significantly decreased from baseline at various post-baseline time points, with statistically significant differences observed at 12 months (*p* < 0.05) and 24 months (*p* < 0.05) after SLIT [114]. An open-label randomized controlled trial demonstrated that SLIT reduced the mean total SCORAD score spanning from baseline to 3 months, and this effect persisted until 12 months (all *p* < 0.05) [115]. In a multicenter, randomized, double-blind, placebo-controlled study, lower SCORAD scores and a greater decrease in skin lesions were observed in the high- and medium-dose *D. farinae* drops groups compared to the placebo group in the full analysis set (FAS) at 36 weeks (*p* < 0.05) [116]. Hajdu et al. conducted a study indicating a significant improvement in SCORAD and objective SCORAD (OSCORAD) after 6 months of SLIT treatment, while the control group showed no significant changes in the assessed values [117]. In a clinical randomized controlled trial, the 2-year, 3-year, and total efficacy rates were significantly higher in the SLIT group than in the control group (*p* < 0.05). The SCORAD score in the SLIT group was significantly lower than that in the control group after AIT, and the SCORAD score in each treatment group at 1-year post-treatment completion was also significantly lower than the corresponding baseline SCORAD score, both exhibiting a *p*-value < 0.05 [118]. A randomized, double-blind, placebo-controlled trial revealed that after 18 months of SLIT for the treatment of AD, there were significant decreases in the SCORAD score and O-SCORAD from baseline in the HDM SLIT group in comparison to the placebo group [119]. Nahm et al. revealed that after SCIT in AD, a significant decrease in the SCORAD values was noted in the patients with mild-to-moderate and severe AD at 12 months compared with baseline. There was also a significant difference in the mean percentages of the decrease in the SCORAD values and the proportion of patients showing a decrease in the SCORAD values at 12 months, compared to baseline, between the patients with severe AD and those with mild to moderate AD (*p* < 0.05) [120].

Zhou et al. reported that SCIT significantly reduced the mean SCORAD score after 3 years of treatment, irrespective of the severity of SCORAD (all *p* < 0.05). The mean reduction ratio of the SCORAD scores was also higher in the SCIT group than in the non-SCIT group, particularly in the cases with moderate and severe SCORAD scores (all *p* < 0.05) [121]. In a 12-month randomized controlled study, after SLIT, the treatment group exhibited a total efficacy rate of 77.78%, which was significantly higher than the 53.85% rate observed in the control group (*p* < 0.05) [122]. In a study involving adults, a significant improvement in EASI (Eczema Area and Severity Index) was observed in the AIT group compared to the placebo group after SCIT [123].

#### 4.1.2. Change in the Daily Rescue Medication Scores after AIT in Patients with AD

The use of other medications such as topical/systemic corticosteroids, calcineurin inhibitors, and oral antihistamines is often necessary for symptom control during the intervention period of AIT [83]. A change in the daily drug score indicated the effectiveness of the AIT treatment. According to Yu et al., the scores for the average daily rescue medications were significantly lower after SLIT than in the control group from 12 months onward (all *p* < 0.05) [114]. In a randomized controlled trial, the mean medication scores significantly decreased 3–12 months after SLIT [115]. In a multicenter, randomized, double-blind, placebo-controlled study, the use of glucocorticoids increased in both the placebo and low-dose SLIT groups. However, the use of glucocorticoids remained relatively low in the high- and medium-dose SLIT groups [116]. In a 12-month randomized controlled study, the average daily drug scores were significantly lower in the treatment group after SLIT in comparison to the control group after SLIT at both 6 and 12 months. Additionally, at the 12-month follow-up, compared with the first month, the decrease in average daily drug scores in the treatment group was significantly greater than that in the control group (both *p* < 0.05) [122]. In a SCIT study involving adults, the medication score significantly decreased in the AIT group 12 months after treatment [123].

#### 4.1.3. Alteration of Visual Analog Scale (VAS) Score after AIT for the Treatment of AD

The visual analog scale (VAS), which ranges from 0 to 10, is a useful index for assessing the symptoms of AD and can be indicative of therapeutic effects [124]. According to Yu et al., the VAS score significantly decreased in the SLIT group compared with the baseline (*p* < 0.05) and control group (*p* < 0.05) from 12 months [114]. Zhou et al. reported that SCIT significantly reduced the pruritus VAS scores from baseline to 0.5 years, with this effect persisting until 3 years. Moreover, the mean reduction ratio of the pruritus VAS scores was higher in the SCIT group than in the non-SCIT group (all *p* < 0.05) [121]. In a 12-month randomized controlled study, it was observed that the VAS scores significantly decreased in the group receiving SLIT compared to those in the control group at the end of the treatment period (*p* < 0.05) [122].

#### 4.1.4. Variation in Biomarkers after AIT for the Treatment of AD

AIT can reduce the levels of allergen-specific IgE antibodies and increase the levels of immunoglobulin G4 (IgG4) antibodies, which is a non-inflammatory immunoglobulin isotype [75]. Similar results have been observed in AIT for allergic rhinitis [125]. A decrease in the absolute eosinophil count (AEC) in peripheral blood was also observed following AIT [126]. These changes are likely to occur in patients with AD following AIT. According to Yu et al., there were no significant differences in serum IgE (sIgE) between the SLIT and control groups at baseline and after 24 months (*p* > 0.05) [114]. In an open-label randomized controlled trial, the levels of *D. farinae*-specific IgG4 increased significantly from baseline to 12 months in the SLIT group (*p* = 0.012). Additionally, the mean IgG4 levels specific to both *D. farinae* and *D. pteronyssinus* at 12 months were significantly higher in the SLIT group than in the control group. However, there were no significant changes in the mean sIgE levels between baseline and 12 months in either the SLIT or the control groups [115]. Nahm et al. demonstrated that the sIgE concentrations and peripheral blood total eosinophil counts significantly decreased from baseline to 12 months, particularly in severe AD (all *p* < 0.05) [120]. Zhou et al. reported that after 3 years of SCIT, the eosinophil counts significantly differed in the SCIT group and complete response (CR) group (SCORAD reduction ratio ≥ 90%) but not in the non-CR group from baseline to 3 years. However, there was no significant difference in the serum total IgE (t-IgE) values between the CR and non-CR groups [121]. In a 12-month randomized controlled study, significantly higher levels of serum sIgG4 were observed in the SLIT treatment group than in the control group at both 6 and 12 months (*p* < 0.05) [122]. In a study involving adults, there was a decrease in the sIgE levels during SCIT therapy, and the IgG4 levels increased after 12 months of immunotherapy in the study group [123]. These findings consistently indicate higher IgG4 levels and lower AEC after AIT. However, it is worth noting that the IgE levels did not show a significant decrease in all the studies.

## 5. Future Prospect of AIT for the Treatment of AD

In the recent guidelines for the treatment of AD, AIT is considered an option but may not be included in all guidelines [109]. The allergen type, appropriate allergen dosages, given intervals, and time of follow-up of AIT are variables in these studies. *D. pteronyssinus* (Der p) and *D. farinae* (Der f) are allergens commonly used in AIT to treat AD. The duration of the course and follow-up intervals in AIT for AD can vary, with studies reporting a range of 6–36 months (Table 3). Higher IgG4 levels and a lower AEC were observed after AIT in our review. In contrast, patients with AD treated with dupilumab exhibited elevated AEC. Moreover, individuals facing dupilumab-related ocular surface disease or facial redness dermatitis showed a notable increase in the AEC [127]. Therefore, additional biomarkers related to the clinical efficacy of AIT for the treatment of AD could be explored. These may include markers such as IgA, IL-10, TGF-B, cluster of differentiation (CD)63, or CD203c [128,129]. Additional clinical studies are essential to enhance our understanding of therapeutic processes, guidelines, efficacy, and practicality. Furthermore, evaluating a broader spectrum of biomarkers may yield a more comprehensive understanding of the immunological changes and responses associated with AIT for the treatment of AD.

## 6. Conclusions

AIT has demonstrated significant therapeutic efficacy for the treatment of AD, attributable to its capability to alleviate inflammation and modulate immune responses. In our comprehensive review, we delved into clinical studies conducted over the past decade concerning AIT for the treatment of AD. We categorized these studies according to severity, daily drug scores, VAS scores, and biomarkers. These studies consistently revealed a reduction in SCORAD and EASI scores, signifying a decrease in AD severity. Furthermore, there was a reduction in daily medication use and lower VAS scores, indicating improved symptom control. In the context of immunological biomarkers, there was consistent evidence indicating elevated IgG4 levels and reduced eosinophil counts after AIT. The infrequent occurrence of severe systemic adverse effects highlighted the safety of this treatment. Exploring the optimal therapeutic processes and identifying efficiency-associated biomarkers are crucial aspects that warrant further investigation. These research directions hold considerable promise for AIT as a therapeutic approach against AD.

## Figures and Tables

**Table 1 ijms-25-01316-t001:** Hanifin and Rajka criteria for diagnosis of AD.

Major Criteria (3 or More Required)	Minor Criteria (3 or More Required)
1. Pruritus; 2. Typical morphology and distribution; ■ Facial and extensor involvement in infancy and children; ■ Flexural lichenification in adults; 3. Chronic or chronically relapsing dermatitis; 4. Personal or family history of atopic disease (asthma, allergic rhinitis, Atopic dermatitis).	1. Xerosis; 2. Ichthyosis, hyperlinear palms, or keratosis pilaris; 3. Immediate skin test reactivity; 4. Raised serum IgE; 5. Early age of onset; 6. Tendency for cutaneous infections; 7. Tendency toward nonspecific hand or foot dermatitis; 8. Nipple eczema; 9. Cheilitis; 10. Recurrent conjunctivitis; 11. Dennie–Morgan infraorbital folds; 12. Keratoconus; 13. Anterior subscapsular cataracts; 14. Orbital darkening; 15. Facial pallor or facial erythema; 16. Pityriasis alba; 17. Anterior neck folds; 18. Pruritus when sweating; 19. Intolerance to wool and lipid solvents; 20. Perifollicular accentuation; 21. Food intolerance; 22. Course influenced by environmental and/or emotional factors; 23. White dermatographism or delayed blanch to cholinergic agent.

**Table 2 ijms-25-01316-t002:** Severity of AD.

Organ System	Manifestations of AD
Mild	Only mild eruptions are observed irrespective of the area.
Moderate	Eruptions with severe inflammations are observed on less than 10% of the body surface area
Severe	Eruptions with severe inflammation are observed on 10% to <30% of the body surface area.
Most severe	Eruptions with severe inflammation are observed on 30% of the body surface area.

Mild eruption: Lesions are seen chiefly with mild erythema, dry skin, or desquamation. Eruption with severe inflammation: Lesions with erythema, papule, erosion, infiltration, or lichenification.

**Table 3 ijms-25-01316-t003:** Exploring clinical study characteristics in AIT for AD patients.

	Study (Years)	Study Design	Number of Patients	Age (Years) (Mean ± SD)	% Men	Time of Evaluation	Course	Severity (Based on SCORAD Score)	Route of AIT	Allergen	Effectiveness of AIT	Change in Rescue Medication Scores	Change in VAS	Change in Biomarkers	Study Summary	Country
1	Yu (2021) [114]	Randomized controlled trial	77 (SLIT: 39; control: 38)	26.5 ± 4.5	45	0, 6, 12, 24 months	2 years	Mild to moderate	SLIT	HDM (Der f)	**Compare to baseline:** ↓SCORAD at month 12, 24	**Compare to control:** ↓at Month 12 to 24	**Compare to baseline and control:** ↓at Month 12 to 24	**Compare to control:** No significant change in IgE	Significantly improved the clinical symptoms and reduced drug use in mild–moderate AD	China
2	Kim (2022) [115]	Open-label, controlled, randomized trial, no placebo	60 (SLIT: 30; control: 30)	8.8 ± 2.7	51.7	3, 6, 9, 12 months	1 year	Mild to severe	SLIT	HDM (Der f, Der p)	**Compare to baseline:** ↓SCORAD at Month 3, 6, 9, 12	**Compare to baseline:** ↓at Month 3, 6, 9, 12	X	**Compare to baseline and control:** ↑IgG4 at month 12 **Compare to control:** No significant changes in IgE at month 12	Improved AD severity	Republic of Korea
3	Liu (2019) [116]	Clinical phase II, multi-center, randomized, double-blind, and placebo-controlled trial	236 (high dose: 60; moderate dose: 55; low dose: 54; control: 57)	31.5 ± 10.8	48.2	0, 4, 10, 16, 24, 36 weeks	36 weeks	Mild to moderate	SLIT	HDM(Der f)	**Compare to control:** ↓SCORAD in week 16 and 36 (in medium- and high-dose allergen)	Low steroid use in high SLIT group	X	X	Improved AD severity and reduced drug use	China
4	Hajdu (2020) [117]	Randomized controlled trial	14 (SLIT: 8; control: 6)	19.0 ± 8.3	50.0	0, 6 months	6 months	Mild to moderate	SLIT	HDM (Staloral) (Der p)	**Compare to control:** ↓SCORAD at month 6	X	X	X	Improved the clinical symptoms and permeability barrier functions	Hungary
5	Huang (2022) [118]	Randomized controlled trial	440 (SLIT: 309; control: 131)	7.3 ± 2.6	61.6	0, 6, 12, 24, 36 months	36 months	Not mentioned	SLIT	HDM (Der p)	**Compare to baseline:** ↓SCORAD in year 1 **Compare to control:**↓SCORAD in year 1, 2, 3	X	X	X	Improved AD severity in children	China
6	Langer (2022) [119]	Randomized, double-blind, placebo-controlled trial	66 (SLIT: 31; control: 35)	19.6 ± 14.3	28.8	0, 3, 6, 9, 12, 15, 18 months	18 months	Mild to severe	SLIT	HDM (Der p)	**Compare to control:** ↓SCORAD at months 18	X	X	X	Improved AD severity	Brazil
7	Nahm (2016) [120]	Observational cohort study	251 (mild: 34; moderate: 123; severe: 94)	19.9 ± 10.1	59.0	0, 12	12 months	Mild to severe	SCIT	HDM (Der f, Der p)	**Compare to baseline:** ↓SCORAD at month 12 (mild to moderate and severe group) **Compare to control:** ↓SCORAD at month 12 (severe group compared to mild to moderate group)	X	X	**Compare to baseline: ** ↓ IgE and peripheral blood total eosinophil counts at month 12		Republic of Korea
8	Zhou (2021) [121]	Retrospective analysis	378 (SCIT: 164; control: 214)	15.5	52.6	0, 6, 12, 24, 36 months	36 months	Mild to severe	SCIT	HDM (Der f, Der p)	**Compare to baseline:** ↓SCORAD at month 6, 12, 24, 36 **Compare to control:** ↓SCORAD at month 36	X	**Compare to baseline:** ↓in year 0.5, 1, 2, 3 **Compare to control:** ↓in year 3	**Compare to baseline:** ↓ eosinophil counts in year 3 (SCIT group and CR group), no significant changes in IgE in year 3 (SCIT group, CR group, and non-CR group)	Significantly reduced the severity and pruritus of moderate to severe AD	China
9	Qin (2014) [122]	Randomized controlled trial	107 (SLIT: 58; control: 49)	27.3 ± 8.2	58.9	0, 1, 6, 12 months	12 months	Not mentioned	SLIT	HDM (Der f)	**Compare to control:** ↑total efficacy rate (77.78% > 53.85%)	**Compare to control:** ↓at month, 6, 12	**Compare to control:** ↓at month 12	**Compare to control:** ↓ IgE and↑IgG4 at month 12	Improved AD severity and reduced drug use	China
10	Bogacz-Piaseczyńska (2022) [123]	Randomised, placebo-controlled, double-blind trial	37 (SCIT: 21; control: 16)	19.2	51.4	0, 12 months	12 months	Moderate to severe (EASI)	SCIT	HDM (Der f, Der p)	**Compare to baseline:** ↓EASI at month 12	**Compare to baseline:** ↓at month 12	X	**Compare to control:** ↑IgG4 at month 6, 12	Improved AD severity	Poland

## Data Availability

Not applicable.

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
