# Peer review of "Advancements in Allergen Immunotherapy for the Treatment of Atopic Dermatitis"

_ijms, 2024, doi:10.3390/ijms25021316_

Round 1
Reviewer 1 Report
Comments and Suggestions for Authors
The Authors elegantly review the potential of AIT in AD.
The manuscript is well-written.
I would encourage the Authors to mention/comment on the different dynamics of eosinophils in AD patients when treated with dupilumab as compared with AIT.
Please see and cite:
Ferrucci S, Angileri L, Tavecchio S, et al. Elevation of peripheral blood eosinophils during dupilumab treatment for atopic dermatitis is associated with baseline comorbidities and development of facial redness dermatitis and ocular surface disease. J Dermatolog Treat. 2022;33(5):2587-2592. doi:10.1080/09546634.2022.2049588
Reviewer 2 Report
Comments and Suggestions for Authors
Guo et al. have provided comprehensive review of the insights for future research clinical practice, exploring allergy immunotherapy (AIT) as a viable option for the management of AD. The review article is well-written, well organized and comprehensively described. Some minor suggestions are as follows:
- English proof editing is for some typos is recommended.
- The Authors are referred to a recently published article related to the antioxidants as a potential therapeutic strategy in the treatment of AD: https://doi.org/10.3390/antiox12101875
- The abbreviations when first introduced should be used consistently thereafter.
- References 36 and 37 are the same.
Comments on the Quality of English Language
English proof editing is for some typos is recommended English proof editing is for some typos is recommended.
Reviewer 3 Report
Comments and Suggestions for Authors
The authors present a review article that aims to explore advancements in allergen immunotherapy for the treatment of atopic dermatitis.
Comments.
1.
If this is a systematic review article, it is suggested to follow the PRISMA guidelines.
2.
The presentation of Table 3 should be reconsidered, including a reevaluation of the study authors (consider omitting the study title), research design (specifying whether it is a clinical trial, double-blind randomized assignment, or other study design), efficacy or effect size, and explanations of study conclusions. If appropriate, it is suggested to consider conducting a meta-analysis.
3.
Perhaps a summary table could assist readers in navigating the essential information within this review article.
Round 2
Reviewer 1 Report
Comments and Suggestions for Authors
No other comments from me.
Reviewer 3 Report
Comments and Suggestions for Authors
No further comment